# Dihydromyricetin Attenuates High-Intensity Exercise-Induced Intestinal Barrier Dysfunction Associated with the Modulation of the Phenotype of Intestinal Intraepithelial Lymphocytes

**DOI:** 10.3390/ijms24010221

**Published:** 2022-12-23

**Authors:** Pengfei Hou, Dawei Wang, Hedong Lang, Yu Yao, Jie Zhou, Min Zhou, Jundong Zhu, Long Yi, Mantian Mi

**Affiliations:** 1Institute of Military Preventive Medicine, Third Military Medical University, Chongqing 400038, China; 2Research Center for Nutrition and Food Safety, Chongqing Key Laboratory of Nutrition and Food Safety, Institute of Military Preventive Medicine, Third Military Medical University, Chongqing 400038, China

**Keywords:** dihydromyricetin, intestinal barrier function, intestinal intraepithelial lymphocytes, high-intensity exercise, aryl hydrocarbon receptor

## Abstract

Background: Exercise-induced gastrointestinal syndrome (GIS) has symptoms commonly induced by strenuous sports. The study aimed to determine the effect of dihydromyricetin (DHM) administration on high-intensity exercise (HIE)-induced intestinal barrier dysfunction and the underlying mechanism involved with intestinal intraepithelial lymphocytes (IELs). Methods: The HIE model was established with male C57BL/6 mice using a motorized treadmill for 2 weeks, and DHM was given once a day by oral gavage. After being sacrificed, the small intestines of the mice were removed immediately. Results: We found that DHM administration significantly suppressed HIE-induced intestinal inflammation, improved intestinal barrier integrity, and inhibited a HIE-induced increase in the number of IELs and the frequency of CD8αα^+^ IELs. Meanwhile, several markers associated with the activation, gut homing and immune functions of CD8αα^+^ IELs were regulated by DHM. Mechanistically, luciferase reporter assay and molecular docking assay showed DHM could activate the aryl hydrocarbon receptor (AhR). Conclusions: These data indicate that DHM exerts a preventive effect against HIE-induced intestinal barrier dysfunction, which is associated with the modulation of the quantity and phenotype of IELs in the small intestine. The findings provide a foundation to identify novel preventive strategies based on DHM supplementation for HIE-induced GIS.

## 1. Introduction

Exercise-induced gastrointestinal syndrome (GIS) such as diarrhea, cramping, nausea, vomiting and bloating are common in endurance athletes, particularly runners and triathletes [1]. The prevalence of exercise-induced GIS is from 30% to 90%, which has negative impacts on the athletic performance and the scores of competitions [2]. There are various studies disclosing the pathophysiological mechanisms, which includes gut microbial dysbiosis, ischemia-reperfusion and metabolic dysfunction. The physiological and pathological factors could contribute to the dysfunction of intestinal barrier and exercise-induced GIS [3,4]. However, the cellular and molecular mechanism as well as the protective strategy of HIE-induced intestinal barrier dysfunction are not clarified.

The integrated intestine barrier can prevent the translocation of harmful substances such as lipopoly saccharide (LPS) from the lumen to the bloodstream. The physical barrier is maintained by intestinal epithelial cells, which are connected to each other by tight junction (TJ) proteins such as claudins, occludin, and zona occludens (ZO-1, ZO-2, and ZO-3). Thus, the improvement of the physical barrier is a vital strategy for the prevention of HIE-induced GIS. However, in recent years, various studies found that intestinal intraepithelial lymphocytes (IELs), which reside between the intestinal epithelial cells (IECs), are a large and diverse population of lymphoid cells involved in tissue homeostasis, epithelial repair and immune response against commensal bacteria and infections [5]. Distributed along the length of the small intestine at a density of 1 per 10 epithelial cells, approximately 90% of IELs express T cell receptors (TCRs) and can be further divided into the natural and induced IELs expressing several cytokines such as interferon-γ (IFN-γ), interleukin-10 (IL-10), tumor necrosis factor-a (TNF-α), keratinocyte growth factor (KGF) and transforming growth factor-β (TGF-β) [6]. Particularly, the pro-inflammatory cytokine IFN-γ promotes TJ dysregulation and enhances intestinal permeability, whereas the anti-inflammatory cytokine IL-10 plays a vital role in maintaining intestinal integrity. Previous studies showed that the interaction between IELs and IECs regulates the mucosal barrier integrity through the cytokines IFN-γ and IL-10 under homeostatic conditions or exogenous insults, while the dysregulated IELs plays an important role in the pathological changes in inflammatory bowel disease (IBD) or celiac disease with an increased IFN-γ and decreased IL-10. For instance, an increased population of TCRγδ-positive IELs (TCRγδ^+^ IELs) was observed in several inflammatory intestinal diseases [7]. These results raise the question that whether IELs would become a critical regulator of the intestinal barrier function and inflammatory response in HIE-induced GIS.

Currently, nutritional supplements have been proven to be effective and important strategies for the prevention of exercise-induced GIS in elite athletes, particularly the antioxidants and anti-inflammatory agents [8]. Certain nutritional agents such as dietary flavonoids could exert anti-oxidative and anti-inflammatory activity, contributing to the prevention of the exercise-induced epithelial damage, endotoxaemia and gastrointestinal syndrome through the decreased level of free radicals [9]. Dihydromyricetin (DHM) is known to have anti-inflammatory and anti-oxidant benefits [10]. In our previous studies, DHM improved physical performance under simulated high-altitude conditions via protecting mitochondrial biogenesis and modulating mitochondrial dynamics in skeletal muscle cells [11]. However, no study reported the effects of DHM on HIE-induced intestinal barrier dysfunction. More and more studies have confirmed that the aryl hydrocarbon receptor (AhR) is a crucial regulator in maintaining the intestinal barrier integrity [12]. The AhR deficiency or the lack of AhR ligands would heighten immune imbalance and increase epithelial damage as well as the intestinal barrier dysfunction, yet AhR activation significantly ameliorated the intestinal permeability by rectifying the expression of the epithelial TJ protein zona-occludens-1 (ZO-1) [13,14]. Thus, AhR might be closely associated with the maintenance of intestinal barrier homeostasis during HIE. Meanwhile, according to the reports, dietary phytochemicals exhibit substantial cell context-dependent AhR agonist as well as antagonist activities. Therefore, as a bioactive flavonoid isolated from Ampelopsis grossedentata, we aim to identify whether DHM supplementation could attenuate HIE-induced intestinal mucosal barrier dysfunction and to reveal the involvement of IELs in DHM-induced benefits.

## 2. Results

### 2.1. DHM Attenuates the Intestinal Inflammation in Mice Induced by High-Intensity Exercise

The indicated two-week training program was established according to the previous study (Figure 1A) [15]. There were no significant changes in the body weight and food intake among the mice in different groups (Figure 1B,C). Compared to the SED group, subjected to HIE obviously increased the inflammatory cells infiltration in the sub-mucosa and the small intestinal villi structure disorder assessed by the HE assay (Figure 1D). Interestingly, the administration of 100 mg/kg·bw/day of DHM for two weeks significantly attenuated HIE-induced intestinal inflammation and villi structure disorder in the small intestine, respectively (Figure 1D). In addition, DHM administration reversed the HIE-induced increase in mRNA expression levels of IFN-γ (Figure 1E) and tumor necrosis factor-α (TNF-α) (Figure 1F) in the small intestine of mice, respectively. Additionally, the mRNA expression of IL-10 in the small intestine was remarkably increased in the DHM-supplemented group compared to the HIE group (Figure 1G). Taken together, our results suggest that DHM administration can significantly suppress HIE-induced intestinal inflammation in mice.

### 2.2. DHM Maintains the Intestinal Barrier Integrity and Reduces Endotoxemia in Mice following High-Intensity Exercise

The intestinal physical barrier is maintained by the intestinal epithelial cells, which are connected to each other by TJ proteins and adhesion molecules. To reveal whether DHM could further ameliorate the intestinal barrier integrity, we tested the endotoxemia, intestinal injury markers as well as the protein and mRNA expressions of the associated TJ proteins in the small intestine of mice. As expected, significantly decreased plasma LPS and I-FABP levels were observed in mice with DHM administration, compared to the HIE-treated mice, implying a reduced endotoxemia resulted from the DHM supplement (Figure 2A,B). In addition, DHM-fed mice showed up-regulated mRNA levels of ZO-1 and occludin (Figure 2C,D) as well as protein levels of occludin and claudin-1 (Figure 2E–G) in the small intestine, respectively, indicating an increased intestinal mucosal integrity aided by DHM administration. Observed by immunofluorescence staining, the expression of E-cadherin was increased in the small intestine of mice from the DHM-administration group compared to the HIE group (Figure 2H,I), revealing that a disrupted intestinal epithelial adherence junction in the small intestine of mice followed by HIE could be reversed by DHM. Collectively, these results demonstrate that DHM improves the HIE-induced intestinal barrier dysfunction and endotoxemia in mice.

### 2.3. DHM Reduces the Number of IELs and the Frequency of CD8αα^+^ IELs in Mice following High-Intensity Exercise

We further detected whether the amelioration of intestinal barrier integrity followed by DHM administration was associated with certain intestinal immune cells. The IELs were isolated from the small intestine in different groups and the phenotypes were detected by flow cytometry (Figure 3A,B). The frequency of CD8αα^+^ IELs (natural IELs) was significantly increased while that of CD8αβ^+^ IELs (induced IELs) was decreased in the HIE group compared with the SED group, respectively (Figure 3C). However, DHM administration induced a decreased percentage of CD8αα^+^ IELs compared to the HIE group, while the percentage of CD8αβ^+^ IELs was not notably affected (Figure 3C). Furthermore, we analyzed the frequency of different subsets of CD8αα^+^ IELs among different groups. Compared to the SED group, there was a significant increase in the frequency of CD8αα^+^TCRγδ^+^ IELs and decrease in CD8αα^+^TCRαβ^+^ IELs in the HIE group, which were dominantly inhibited by DHM administration (Figure 3D). Likewise, there was a similar trend in the frequency of total TCRγδ^+^ and TCRαβ^+^ IELs in the small intestine of mice after DHM administration (Figure 3E). Moreover, we found the number of IELs in the small intestine was increased after HIE treatment, which was inhibited by DHM administration (Figure 3F). To reveal the underlying mechanisms that DHM reduced the infiltration of CD8αα^+^ IELs in the intestine, we examined the mRNA expressions of several critical genes in the intestinal tissue. The mRNA expressions of IL-15 and IL-7 as well as CD132, which were responsible for the development and maintenance of CD8αα^+^ IELs, were increased significantly in HIE group. However, DHM treatment significantly suppressed the mRNA expressions of the indicated genes compared to the HIE group (Figure 3G–I). Collectively, these results implied that HIE-induced intestinal barrier dysfunction might be related with the infiltration of natural IELs in the small intestine, and DHM-induced amelioration of the intestinal inflammation and barrier integrity might be associated with the suppression of IELs quantity and the frequency of CD8αα^+^ IELs in the small intestine.

### 2.4. DHM Modulates the Immune Function of CD8αα^+^ IELs in Mice Following High-Intensity Exercise

We guess that the ameliorative effect of DHM on HIE-induced intestinal barrier dysfunction might be associated with the immune function changes in CD8αα^+^ IELs in the small intestine. We further investigated the activation and migration markers as well as the cytokines expression of CD8αα^+^ IELs (natural IELs) in the small intestine of mice. As expected, the expressions of the critical cytokines of IFN-γ and IL-10 in the CD8αα^+^ IELs were measured by intracellular staining and flow cytometry. As shown in Figure 4A–D, there was a significant reduction in IL-10 and increase in IFN-γ of CD8αα^+^ IELs from the HIE group, while these effects were markedly reversed by DHM administration. Additionally, HIE resulted in a decreased mRNA expression of KGF-1 in the small intestine of mice, a protective cytokine for intestinal barrier. However, the altered mRNA expressions of KGF-1 were rescued by DHA administration (Figure 4E). In addition, CD103, which is associated with gut homing and retention of CD8αα^+^ IELs, was found to be increased in the CD8αα^+^ IELs from the HIE mice (Figure 4F,G). Additionally, a significant increase in the expression of CD69, an early T cell activation marker in the HIE group was also observed, implying the activation of natural IELs (Figure 4H,I). Interestingly, the expressions of CD69 and CD103 were suppressed in the HIE + DHM group compared to the HIE mice (Figure 4F–I). Additionally, there was a similar mRNA expression trend of CD103 in the small intestine of mice after DHM administration (Figure 4J). Taken together, our results demonstrate that the activation and migration-related markers of IELs as well as certain critical cytokines secreted by CD8αα^+^ IELs were remarkably changed, implying that the immune function of CD8αα^+^ IELs in the intestine of mice was notably modulated by DHM supplementation.

### 2.5. DHM Attenuates HIE-Induced Intestinal Barrier Dysfunction Associated with the Activation of AhR

It has been demonstrated that the activation of AhR plays an important role in the intestinal barrier integrity, and it could be regulated by some special microbial metabolites and dietary AhR ligands. Thus, we quantified the mRNA expressions of AhR and CYP1A1, a hallmark of AhR activation. DHM administration could stimulate AhR, characterized by the increased expression of CYP1A1, while the mRNA level of AhR in the small intestine was not notably affected by DHM administration (Figure 5A,B). Next, to get more insight into the underlying mechanism involved with the effect of DHM on IELs, we performed a virtual docking validation to examine the potential interactions between DHM and AhR using a AutoDoc vina software (The Scripps Research Institute, La Jolla, CA, USA), which could further show the binding conformation and the detailed interaction information. We found that DHM formed a hydrophobic interaction with the residue Ala129 and hydrogen-bond interactions with the residues Tyr157, Trp174, Pro178 and Tyr239 in the crystal structure of AhR (Figure 5C). To further confirm it in vitro, HEK-293 cells were treated with DHM, and there were no obvious changes in the cell viability when treated with DHM at a concentration of lower than 40 μM for 24 h (Figure 5D). In addition, the mRNA expression of CYP1A1 was notably up-regulated by the treatment of 20 μM DHM and 5nM FICZ (a reported AhR agonist), indicating that DHM could activate AhR in HEK-293 cells (Figure 5E). Furthermore, a highly sensitive AhR-responsive assay (Denison et al., 1998) was applied to confirm the modulation of DHM on the translocation of AhR, in which the HEK-293 cells were transfected with a construct containing multiple DRE promoter elements. The results showed that 20 µM of DHM could activate the luciferase activity in HEK-293 cells, suggesting that DHM could increase the nuclear translocation of AhR (Figure 5F). Moreover, the immunofluorescence staining showed that AhR was expressed in both the cytoplasm and nucleus of HEK-293 cells without DHM treatment, while supplementation with DHM or FICZ led to an increased translocation of AhR into the nucleus (Figure 5G). Taken together, DHM attenuates HIE-induced intestinal barrier dysfunction, which might be associated with the activation of AhR.

## 3. Discussion

Gastrointestinal syndrome is common in endurance sports, particularly high-intensity exercises. The primary aim of this study was to investigate the preventive effect of DHM against high-intensity exercise-induced GIS, and to reveal the possible involvement of IELs with DHM-induced benefits. Our results show that DHM, as an AhR agonist, attenuates HIE-induced GIS through amelioration of intestinal barrier dysfunction, which might be associated with the modulation of the quantity and phenotype of IELs and the physical barrier. Additionally, our findings add to the current literature that dietary supplements of DHM might become an effective strategy towards populations suffering from the exercise-induced GIS. Furthermore, IELs, which are located between the intestinal epithelial cells, are critical for the maintenance of the intestinal barrier integrity. Additionally, the results provide novel insights into the mechanism that DHM could modulate the quantity, immune function and the frequency of certain subsets of IELs (in particular CD8αα^+^ IELs) in the intestine of mice followed by high-intensity exercise, which are associated with the beneficial effects of DHM on intestinal dysfunction and GIS (Figure 6).

The gastrointestinal system plays an important role in an athlete’s ability to perform strenuous exercise. Proper gastrointestinal function is important for achieving nutrient intake requirements and endurance performance [15,16]. In contrast, a compromised gastrointestinal system will likely impair performance and post-exercise nutrient absorption and may cause serious gastrointestinal symptoms during prolonged exercise [17,18,19]. The intestinal mucosal barrier is an integrated part of the organ, including the physical, chemical, biological and immunologic barriers, respectively. According to previous animal and human studies, as exercise intensity and duration increases, intestinal inflammation injury, mucosal erosions and TJ proteins dysregulation were more prone to occur, leading to increased intestinal permeability, bacterial translocation and endotoxaemia [20,21,22]. The intestinal barrier integrity was associated with the balance of different inflammatory cytokines. In some physiological and pathophysiological conditions, when the balance of Th1 (IFN-γ, TNF-α, IL-2) and Th2 (IL-4, IL-6, IL-10) cytokines were broken, the intestinal barrier function was largely affected [23]. Recently, several studies have indicated a direct effect of IFN-γ on the distribution of TJ proteins [24]. One study showed that IFN-γ regulated E-cadherin stability by a Fyn kinase-dependent mechanism. In addition, IL-10 was usually considered to be the most important anti-inflammatory cytokine [25]. Exogenous IL-10 administration significantly attenuated total parenteral nutrition (TPN)-associated decline in ZO-1, E-cadherin and occludin expressions, as well as the loss of intestinal barrier function [26]. In our study, HIE induced a significant intestinal barrier injury and dysfunction, as evidenced by increased expressions of inflammatory cytokines such as IFN-γ, TNF-α and decreased IL-10 expression (Figure 1). Moreover, HIE led to a reduced expressions of TJ proteins including ZO-1, occludin, claudin-1 and E-cadherin, suggesting an aggravated intestinal barrier dysfunction was induced by HIE (Figure 2). Moreover, the mechanism of HIE-induced intestinal barrier dysfunction might have resulted from splanchnic hypoperfusion and the associated oxygen deprivation [27]. The splanchnic hypoperfusion would create epithelial injury associated with apical erosion and likely dysfunction of all epithelial cell types such as enterocyte, goblet, Paneth and enteroendocrine cell [22,28]. Additionally, the physical breaks in the epithelium, damaging the multi-protein complex (such as claudins and occludin) of the tight-junction and/or promoting dysfunction to tight-junction regulatory proteins (i.e., zona-occludens) would result in a combined consequence of increased intestinal permeability [29].

A variety of studies have demonstrated that exercise induced considerable physiological or pathological changes in the immune system. The interactions between exercise stress and the immune system provide a unique opportunity to evaluate the role of underlying stress and immunophysiological mechanisms. Recent studies suggested that IELs were crucial for the maintaining of the intestinal mucosal barrier. However, there were few studies focusing on the role of IELs in the intestinal mucosal of mice following HIE stress. IELs can be classified into TCR^+^ and TCR^−^ subsets. TCR^+^ IELs can be further divided into induced (also termed type-a or conventional) and natural (also termed type-b or unconventional) IELs. Type-a IELs express αβTCRs with CD4 or CD8α, similarly to the conventional T cells, which are major histocompatibility complex (MHC) class II and MHC class I restricted, respectively. Type-b IELs do not express either CD4 or CD8β, but they are CD8αα^+^ with TCRγδ^+^ or TCRαβ^+^. There are several markers that are associated with the function of IELs, particularly CD69 and CD103. CD69 is an early T cell activation marker, which can reflect activated IELs [30]. CD103 is associated with the gut homing and retention of IELs [31]. As shown in Figure 3, our research demonstrated that HIE induced an increased number of IELs, in particular, an increased frequency of CD8αα^+^ IELs (natural IELs) and a decreased frequency of CD8αβ^+^ IELs (induced IELs), respectively. Moreover, gated by TCRαβ and TCRγδ, HIE resulted in a higher frequency of CD8αα^+^TCRγδ^+^ IELs and a lower frequency of CD8αα^+^TCRαβ^+^ IELs. Meanwhile, there was an increased CD103 and a decreased CD69 expression in the CD8αα^+^ IELs of mice followed by HIE (Figure 4). For the first time, our findings provided evidence that the quantity and phenotype of IELs might be changed in high-intensity exercises.

Previous studies showed that some dietary supplements may contribute to the prevention and/or attenuation of exercise-induced GIS. DHM is a major flavonoid isolated from Ampelopsis grossedentata, which has been widely used as an anti-inflammatory, anti-oxidative, lipid and blood glucose-lowering agent in China for many years [32,33,34]. In this study, DHM administration notably reduced the expressions of certain inflammatory cytokines as well as increased the TJ and adhesion junction proteins in the intestine of mice following HIE (Figure 1 and Figure 2). Furthermore, DHM suppressed HIE-induced changes in the quantity and frequencies of certain IELs subsets (in particular CD8αα^+^ IELs) and modulated the associated cytokines from IELs (Figure 3). It is clear that IELs play a critical role in the intestinal barrier by producing cytokines, and the activated CD8αα^+^ IELs may produce more pro-inflammatory cytokines such as IFN-γ and less anti-inflammatory factors such as IL-10. Additionally, our research implied that DHM attenuated HIE-induced gut barrier impairment by inhibiting the activation of natural IELs and promoting a transformation from pro-inflammatory to anti-inflammatory phenotypes in vivo (Figure 4). Additionally, the DHM-induced modulation on the phenotype of IELs might be associated with its benefits in prevention against HIE-induced intestinal barrier dysfunction. These results were similar to the anti-inflammatory effects of DHM on LPS-induced RAW2264.7 macrophages in our previous study [25,26]. However, the mechanism that DHM regulated the cytokines secretion from CD8αα^+^ IELs is still unknow. In addition, there remains an unresolved question about the induced IELs (CD8αβ^+^ IELs). Whether CD8αβ^+^ IELs are also essential for the increased susceptibility to opportunistic pathogens in the small intestine and exogenous bacterial infection followed by HIE remains further elucidated [35].

In addition, intestinal cytokines secreted by IECs play a critical role in the regulation of the number and function of IELs such as IL-7 and IL-15. IL-7Rs, consisting of the common cytokine receptor γc (CD132) with the unique IL-7Rα, have been detected on the surface of thymocytes and IELs [36]. Previous studies demonstrated that IL-7 was crucial for the development of TCRγδ^+^ T cells in the intestinal mucosa of mice [37]. Additionally, the maintenance of CD8αα^+^TCRαβ^+^ and TCRγδ^+^ IELs was in an IL-15-dependent manner [38]. The chronic up-regulation of IL-15 in the intestinal mucosa was a hallmark of celiac disease and it was related to the degree of mucosal damage [39]. Recently, studies have found that I/R resulted in the disruption of TJ proteins (claudin-1 and ZO-1), and it could be attenuated by KGF secreted by IELs [40]. In our study, HIE induced a reduction in KGF-1 and increase in IL-15, IL-7 in the small intestine, which were also improved by DHM supplementation (Figure 3). Additionally, the results also implied that DHM could maybe regulate the crosstalk between IECs and IELs, and it may be playing a key role in the cytokines modulation of IECs following the HIE.

AhR is expressed by different cell types in the gut, such as IECs, IELs, Th17 cells, innate lymphoid cells (ILCs), and regulate intestinal barrier function [41]. Upon ligand binding, AhR translocates to the nucleus and binds to its dimerization partner, AhR nuclear translocator (ARNT). The AhR–ARNT complex then activates the transcription of a variety of target genes that contain dioxin response elements (DRE) or xenobiotic response elements (XREs), including cytochrome P450 (CYP)1A1 [42]. CYP1A1 expression was detected for the indication of AhR activation [43]. In both animal models and mammalian cell lines studies, the remarkable potential of dietary-derived AhR agonists, such as cruciferous vegetables and flavonoids, indole derivatives (indole-3-carbinol (I3C), indole-3-acetonitrile (I3AC)), and/or AhR ligand-producing probiotic organisms as preventive and therapeutic interventions available for intestinal pathological conditions [44]. Based on the study, DHM could active the AhR [45]; previously, we also found DHM can ameliorate the progression of liver fibrosis and inhibit HSC activation by inducing autophagy and enhancing NK cell-mediated killing through the AhR-NF-κβ/STAT3-IFN-γ signaling pathway, providing new insights into the preventive role of DHM in liver fibrosis [46]. As shown in Figure 5, our current study first found that DHM is a dietary-derived AhR agonist in a highly sensitive AhR responsive assay. Additionally, the supplementation of DHM could improve the HIE-induced decrease in CYP1A1 mRNA, which implied the amelioration of disrupted intestinal barrier integrity was associated with the activation of AhR. Moreover, a previous study showed that oral administration of the AhR agonist β-naphthoflavone decreased DSS-induced colitis severity and the production of pro-inflammatory cytokines, such as tumor necrosis factor (TNF)-α, IL-6, and IL-1β [47], which was consistent with our results. The virtual docking validation by AutoDoc vina software showed that there was a potential interaction between DHM and AhR. However, the precision mechanism and direct causal relationship between DHM and AhR will be explained by some new techniques and animal models such as AhR^−/−^ in IECs or IELs of mice.

In conclusion, our results collectively indicate that DHM could improve HIE-induced intestinal dysfunction. Additionally, the quantitative and phenotypic modulation of IELs, particularly the CD8αα^+^ IELs, appear to play a crucial role in the protective effect of DHM on the maintenance of gut barrier integrity and inhibition of gut inflammation of HIE-induced mice. Our research also implied that the beneficial effects induced by DHM might be associated with the AhR activation. These results open a new avenue of research regarding the potential protective effects of DHM on HIE-induced GIS.

## 4. Materials and Methods

### 4.1. Animals and Experimental Protocol

Male C57BL/6 mice, weighing 20–22 g at 8 weeks of age, were housed at a controlled temperature (22–25 °C) and humidity (50–55%). The mice were maintained on a 12 h light/12 h dark cycle in the Experimental Animal Center of Third Military Medical University (Chongqing, China). They were administrated with a chow diet (10% fat, 70% carbohydrate, 20% protein; D12450B, Research Diets, New Brunswick, NJ, USA), while food and water were provided ad libitum. Thirty mice were randomly divided into three groups, including the sedentary group (SED group, *n* = 10), HIE group (*n* = 10) and a group of HIE mice receiving a dietary supplement of DHM (HIE + DHM group, *n* = 10), respectively. DHM was administered by oral gavage once a day for two weeks at a dose of 100 mg/kg of body weight approximately 2 h before exercise. This dosage was chosen following a previous study demonstrating its beneficial effects on exercise endurance [11]. For the first week of acclimatized training, mice in the HIE and HIE + DHM group were placed in a motorized treadmill (SANS Biological Technology, Nanjing, China) in the morning, running at a speed of 15 m/min once a day for 10 min for 5 days, and rest for the remaining 2 days. From day 8, mice in the HIE and HIE + DHM groups ran with the speed of 15 m/min at the beginning for 10 min, and then increased to the speed of 25 m/min by the acceleration of 1 m/min^2^ within the next 10 min. The mice ran for 1 h once a day in the second week at this speed, which was equivalent to 80% VO_2max_ according to the previous study [48]. The mice in the three groups were weighed every two days. After being subjected to HIE, all the mice were sacrificed immediately, and five of the mice in each group were used for analyzing the quantity and phenotype of IELs. Additionally, the serum and small intestine tissues were collected from the remaining five mice in each group and subsequently stored at −80 °C. All animal experiments were approved by the Institutional Animal Care and Use Committees of Third Military Medical University (Chongqing, China; Approval SYXC-2017-0002) and followed the National Research Council Guidelines.

### 4.2. Biochemical Parameters

Serum parameters, including lipopolysaccharide (LPS), intestinal fatty acid binding protein (I-FABP) were quantified using enzymatic assays (mlbio, Shanghai, China) according to the manufacturer’s instructions.

### 4.3. Histological Analysis

The small intestines were dissected and fixed in 4% formaldehyde for at least 24 h. Then, the samples were embedded in paraffin, sectioned at 5 μm and stained with hematoxylin-eosin (HE). The microscopic examination was performed, and photographs were taken using a light microscope (Leica, Wetzlar, Germany).

### 4.4. Immunofluorescence Staining

Segments of small intestines were harvested and fixed in 4% paraformaldehyde. After that, 5 μm-thick, paraffin-embedded sections of colon were prepared and incubated in 0.5% Triton X-100 for 1 h at room temperature. The sections were then blocked with 5% bovine serum albumin (BSA) at room temperature for 30 min and were subsequently incubated with the primary antibodies against E-cadherin (1:100, Proteintech, Rosemont, IL, USA) overnight at 4 °C. Furthermore, the sections were incubated with the goat anti-mouse FITC (1:300, Servicebio, Wuhan, China) antibody at room temperature for 2 h. After being washed with PBS, the sections were incubated with 4′,6-diamidino-2-phenylindole (DAPI) (Beyotime, Nantong, China). Subsequently, they were washed and mounted with Fluoromount-G (SouthernBiotech, Birmingham, AL, USA). The fluorescence was visualized by laser scanning fluorescence microscopy (Leica TCS SP5; Leica). The mean fluorescence intensity was measured using Image-J V 1.8.0 software.

Immunofluorescence microscopy analysis HEK-293 cells were seeded on glass coverslips (NEST, Wuxi, China) in 24-well plates. Following the indicated treatment, the cells were then fixed in 4% paraformaldehyde and followed by permeabilization with 0.25% Triton X-100 (Beyotime, China) for 10 min. Nonspecific binding sites were blocked with 5% goat serum for 1 h. The slides were incubated with a primary antibody against AhR (1:100, Affinity, San Francisco, CA, USA) at 4 °C overnight and then incubated with Alexa Fluor^®^ 488 goat anti-rabbit IgG (H + L) secondary antibody (1:500, Beyotime, China) at 37 °C for 2 h. Nuclei were counterstained with DAPI. Samples were analyzed using laser confocal scanning microscopy (Zeiss, Jena, Germany).

### 4.5. Isolation of IELs and Flow Cytometry Analysis

As for the isolation of IELs, the small intestine was cut open longitudinally followed by the removal of associated fat tissues and Peyer’s patches. Then, the small intestines were washed with iced PBS and shaken in D-Hank’s (Beyotime, China) medium containing 5 mM of ethylenediaminetetraacetic acid (EDTA), 1 mM of dithiothreitol (DTT) and 10 mM of N-2-hydroxyethylpiperazine-N-ethane-sulphonicacid (HEPES) for 15 min at 37 °C three times, and then were homogenized manually through 70-μm cell strainers for collecting the IELs. Mononuclear cells were then harvested from the interlayer between 40% and 70% Percoll (Sigma, St. Louis, MO, USA) gradient after a spin at 900× *g* for 20 min at room temperature. The isolated IELs were stained with antibodies against CD3ε, CD8α, CD8β, TCRαβ and TCRγδ (Biolegend, San Diego, CA, USA) for 20–30 min at 4 °C. For IL-10 and IFN-γ stainings, the IELs were stimulated with 100 ng/mL of phorbol-12-myristate 13-acetate (PMA) and 1 mg/mL of ionomycin using 10 μg/mL of brefeldin A (BD Biosciences, Franklin Lakes, NJ, USA) for 6 h at 37 °C and 5% CO_2_. After stimulation, cells were then fixed and permeabilized using Fixation/Permeabilization kit (BD Biosciences, USA) as directed before intracellular staining using antibodies against IFN-γ and IL-10 (BD Biosciences, USA). Cells were acquired on a BD FCSVerse flow cytometer (BD Biosciences, USA) and analyzed with FlowJoV 10.6.2 software.

### 4.6. Cell Culture and Treatment

HEK-293 cells were cultured in DMEM (Invitrogen, Waltham, MA, USA) supplemented with 10% of fetal bovine serum (HyClone, Logan, UT, USA), 1% glutamine (Beyotime, China) and 1% penicillin-streptomycin (Beyotime, China) in a 5% CO_2_ humidified incubator at 37 °C. Cells from 3rd to 6th passages were used for the following experiments. When growing to 80–90% confluence, cells were treated with different concentrations (10, 20 μM) of DHM (HPLC purity ≥ 98%, Must bio-technology, Chengdu, China) or DMSO (as the control; Sigma, USA) for 24 h.

### 4.7. Cell Viability

Cell viability was analyzed using a Cell Counting Kit-8 (Dojindo, Kumamoto, Japan) following the manufacturer’s instructions. Briefly, 1 × 10^4^ cells were seeded into 96-well microplates and then treated with different concentrations (0, 2.5, 5, 10, 20 and 40 μM) of DHM for 24 h. Subsequently, a cell counting kit CCK-8 was added to the wells (10 μL/well), and the microplate was incubated at 37 °C for 2 h. The absorbance was measured using a monochromator microplate reader (Molecular Devices, Sunnyvale, CA, USA) at 450 nm. Cell viability was calculated from the ratio of the optical density of experimental cells to that of control cells (set as 100%).

### 4.8. Cells Transfection and Luciferase Reporter Assay

HEK-293 cells (2 × 10^4^ cells per well) were seeded in 24-well plates and incubated until the cells reached 60–70% confluence. Cells were then transfected with the Opti-MEM reduced serum medium (Invitrogen, USA) containing pDRE3 (0.5 μg per well) luciferase reporter vector and control vector of pRL-TK (0.025 μg per well) using Lipofectamine 2000 transfection reagent (Invitrogen, USA), following the manufacturer’s instructions. Four hours after transfection, the media were removed, and fresh media containing the DHM or FICZ were added. Cells were incubated for an additional 24 h before harvesting in the reporter lysis buffer. The cell lysates were centrifuged at 12,000× *g*, and the luciferase activity was determined using the Dual Luminescence Assay Kit (Beyotime, China) following the manufacturer’s instructions. The intensity of light emission from assays of cell extracts was determined using a luminometer microplate reader (Molecular Devices, San Jose, CA, USA), as described in the manufacturer’s instructions.

### 4.9. Quantitative Real-Time Polymerase Chain Reaction (qRT-PCR)

Total RNA was extracted from small intestine tissues of mice using TRIzol reagent (Invitrogen Life Technologies, Grand Island, NY, USA). Reverse transcription of mRNA into cDNA was carried out with PrimeScript RT master mix (Takara, Dalian, China). Additionally, PCR assay was performed using a SYBR Premix Ex Taq (Takara, Dalian, China) with qTOWER 2.2 (Analytik, Jena, Germany). Each sample was processed in triplicate and normalized to β-actin with the 2^−ΔΔCT^ method. The prime sequences are listed in Table 1.

### 4.10. Western Blot Analysis

Proteins were extracted from the small intestine by using a tissue lysis buffer (Thermo Scientific, Waltham, MA, USA) with a protease inhibitor cocktail (Roche Diagnostic, Manheim, Germany). Samples were separated in sodium dodecyl sulfate-polyacrylamide gel electrophoresis and transferred to nitrocellulose membranes. The membranes were blocked by 5% dried skimmed milk for 1 h, and were incubated with the corresponding primary antibodies under rotation overnight at 4 °C. The membranes were incubated for 1 h with the peroxidase-conjugated secondary antibody at room temperature, and the proteins were visualized using a chemoluminescence system (Fusion, Paris, France) with Millipore Immobilon ECL substrate (Millipore, Inc., Burlington, MA, USA). The primary antibodies used were Occludin (1:1000, Abcam, Waltham, MA, USA), Claudin-1 (1:1000, Invitrogen, USA) and β-actin (1:1000, Santa Cruz, Dallas, TX, USA), respectively.

### 4.11. Molecular Docking

Docking calculations were carried out using AutoDoc vina software (The Scripps Research Institute, USA). The crystal structure of the protein target AhR (PDB ID: 4m4x) was retrieved from RCSB-PDB (http://www.rcsb.org). The parameters were set to default values.

### 4.12. Statistical Analysis

All analyses were conducted in SPSS 13.0 (Chicago, IL, USA). All experimental data were expressed as mean ± S.E.M. Statistical differences between two groups were determined with unpaired two-tailed Student’s *t*-test. Multiple groups were tested by one-way ANOVA followed with Bonferroni multiple comparison tests. *p*-values less than 0.05 were considered statistically significant.

## Figures and Tables

**Figure 1 ijms-24-00221-f001:**
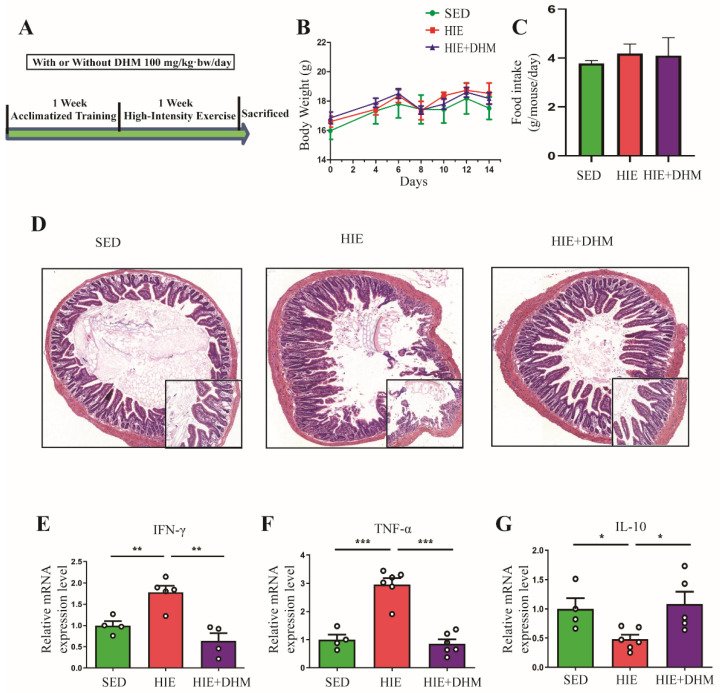
DHM attenuates the intestinal inflammation in mice induced by high-intensity exercise. (**A**) Schematic diagram of the two-week training program and the supplement of DHM. (**B**) Body weight of mice was measured every two days (*n* = 8). (**C**) Food intake of mice in different groups (*n* = 8). (**D**) Representative photographs of small intestine sections with HE staining (50× full graph; 200×, lower right corner) (*n* = 4). (**E**–**G**) The mRNA expression levels of certain pro-inflammatory and anti-inflammatory cytokines IFN-γ (**E**), TNF-α (**F**) and IL-10 (**G**) in small intestine tissues were assessed by qRT-PCR assay (*n* = 4–6). Data were expressed as means ± SEM. The experiments were replicated 2–3 times. The data shown were from one representative experiment, * *p* < 0.05, ** *p* < 0.01 and *** *p* < 0.001. Multiple groups were tested by one-way ANOVA followed by Bonferroni’s post hoc test.

**Figure 2 ijms-24-00221-f002:**
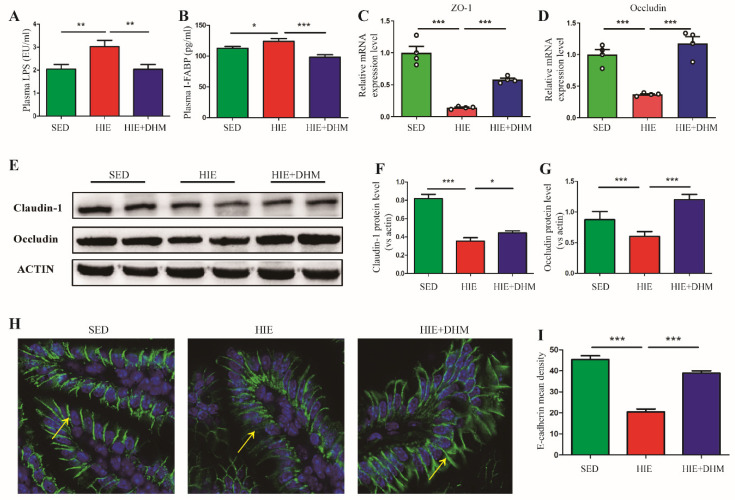
DHM maintains the intestinal barrier integrity and reduces endotoxemia in mice following high-intensity exercise. (**A**) The plasma LPS level was measured by the corresponding assay kit (*n* = 8). (**B**) The plasma IFABP level was measured by the corresponding assay kit (*n* = 6–8). (**C**,**D**) The relative mRNA expression levels of ZO-1 (**C**) and occludin (**D**) in small intestine tissues were detected by qRT-PCR assay (*n* = 4). (**E**) The protein expression levels of claudin-1 and occludin were evaluated by Western blotting. (**F**,**G**) Relative protein levels of claudin-1 (**F**) and occludin (**G**) were quantified by densitometry (*n* = 4). (**H**) Representative images of E-cadherin immunofluorescence staining in the small intestine among the three groups. Green fluorescence represents the expression of E-cadherin protein in the aggregated form (the yellow arrow), while the blue fluorescence with DAPI represents the cell nucleus. (**I**) The bar charts show quantification of fluorescence intensity (*n* = 4). Data were expressed as means ± SEM. The experiments were replicated 2–3 times. The data shown were from one representative experiment, * *p* < 0.05, ** *p* < 0.01 and *** *p* < 0.001. Multiple groups were tested by one-way ANOVA followed by Bonferroni’s post hoc test.

**Figure 3 ijms-24-00221-f003:**
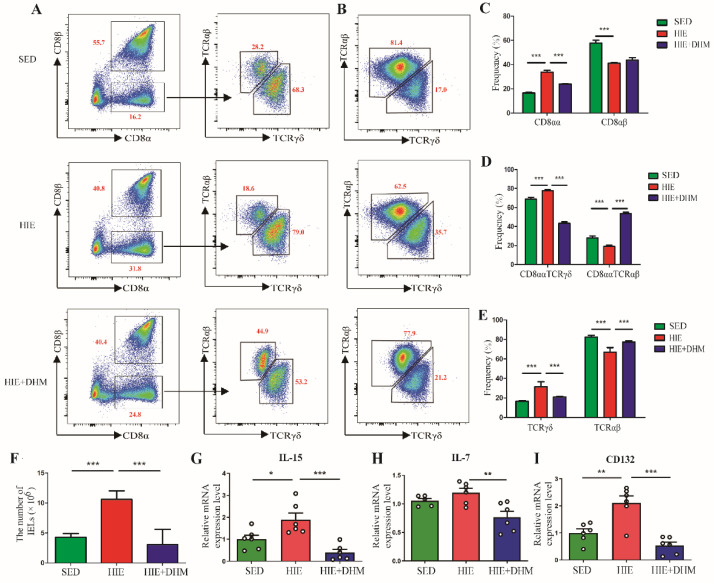
DHM reduces the number of IELs and the frequency of CD8αα^+^ IELs in mice following high-intensity exercise. (**A**,**B**) The IELs from the small intestine of mice were stained as indicated. Cell populations with the frequency of gated cells were shown by different cell phenotype markers (cells were gated by CD3^+^) (*n* = 5). (**C**) The frequencies of CD8αα and CD8αβ on CD3^+^ IELs cell populations among three groups were shown. Cells were stained with anti-CD3, anti-CD8α and anti-CD8β mAbs and positively gated by CD3 (*n* = 5). (**D**) The frequencies of CD8ααTCRαβ and CD8ααTCRγδ on CD3^+^ IELs cell populations among three groups were shown. Cells were stained with anti-CD3, anti-CD8α, and anti-CD8β, anti-TCRαβ, and anti-TCRγδ mAbs and positively gated by CD3 (*n* = 5). (**E**) The frequencies of TCRαβ and TCRγδ on CD3^+^ IELs cell populations among three groups were shown. Cells were stained with anti-CD3, anti-TCRαβ, and anti-TCRγδ mAbs and positively gated by CD3 (*n* = 5). (**F**) The number of IELs in the small intestine among three groups were shown (*n* = 5). (**G**–**I**) The mRNA expression levels of IL-15 (**G**), IL-7 (**H**) and CD132 (**I**) in the small intestine tissues were assessed by qRT-PCR assay (*n* = 5–6). Data were expressed as means ± SEM. The experiments were replicated 2-3 times. The data shown were from one representative experiment, * *p* < 0.05, ** *p* < 0.01 and *** *p* < 0.001. Multiple groups were tested by one-way ANOVA followed by Bonferroni’s post hoc test.

**Figure 4 ijms-24-00221-f004:**
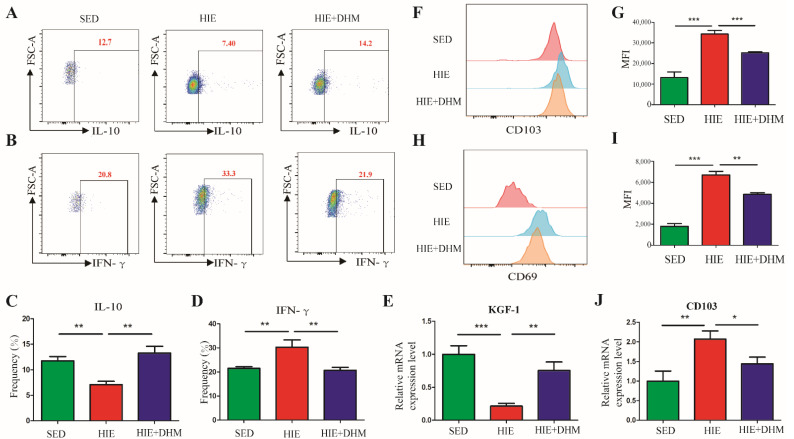
DHM modulates the immune function of CD8αα^+^ IELs in mice following high-intensity exercise. (**A**,**B**) After the IELs were stimulated by PMA and ionomycin with Golgi Stop for 6 h, IL-10 (**A**) and IFN-γ (**B**) expressions in CD8αα^+^ IELs cells were analyzed by flow cytometry (cells were gated by CD3^+^ and CD8α^+^CD8β^−^) (*n* = 4). (**C**,**D**) Frequencies of IL-10-expressing cells and IFN-γ-expressing cells in CD8αα^+^ IELs population in the small intestines were shown (*n* = 4). (**E**) The mRNA expression level of KGF-1 in small intestine tissues were assessed by qRT-PCR assay (*n* = 6). (**F**,**G**) The expression of the surface marker CD103 on CD8αα^+^ IELs of the small intestine in mice was detected by flow cytometry and the mean fluorescence intensity (MFI) among three group mice was shown (cells were gated by CD3^+^ and CD8α^+^CD8β^−^) (*n* = 4). (**H**,**I**) The expression of the surface marker CD69 on CD8αα^+^ IELs in the small intestine of mice was detected by flow cytometry and the mean fluorescence intensity (MFI) among the three groups of mice was shown (cells were gated by CD3^+^ and CD8α^+^CD8β^−^) (*n* = 4–5). (**J**) The mRNA expression level of CD103, an activation and migration marker of CD8αα^+^ IELs in small intestine tissues was assessed by qRT-PCR assay (*n* = 5–6). Data were expressed as means ± SEM. The experiments were replicated 2–3 times. The data shown were from one representative experiment, * *p* < 0.05, ** *p* < 0.01 and *** *p* < 0.001. Multiple groups were tested by one-way ANOVA followed by Bonferroni’s post hoc test.

**Figure 5 ijms-24-00221-f005:**
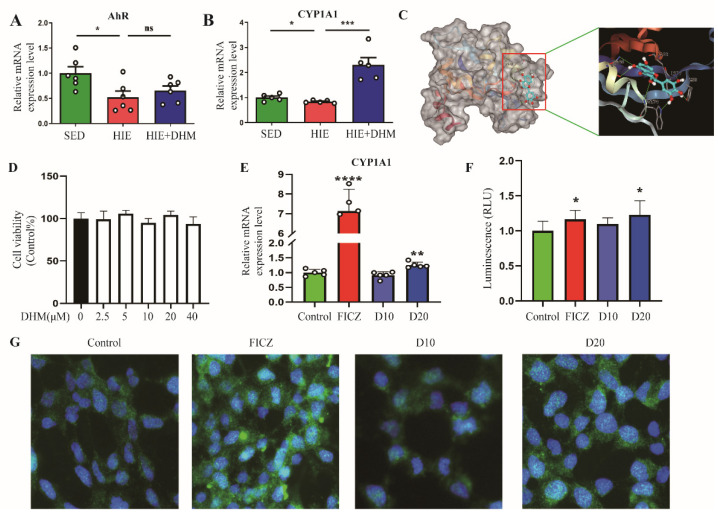
DHM attenuates HIE-induced intestinal barrier dysfunction associated with the activation of AhR. (**A**,**B**) The mRNA expression levels of the CYP1A1 (**A**) and AhR (**B**) in the small intestine tissues were assessed by qRT-PCR assay (*n* = 5–6). (**C**) The binding conformation between DHM and AhR was predicted using the AutoDoc vina software, and DHM was shown as a ball-stick model with color by atoms. The hydrogen-bond interactions were highlighted using green dotted lines while the hydrophobic interactions were highlighted using gray dotted lines. The labeled protein residues were based on a licorice model. (**D**) HEK-293 cells treated with different concentrations of DHM for 24 h. Cell viability was measured with a Cell Counting Kit-8. (**E**) The mRNA expression levels of the CYP1A1 in HEK-293 cells were assessed by a qRT-PCR assay. (**F**) AhR-mediated transactivation in stably transfected HEK-293 cells. Cells were transfected with pDRE3 and treated with DMSO (control), 5 nM of FICZ, and DHM (10 and 20 µM), and the luciferase activity was determined. (**G**) The nuclear localization of AhR (green) followed by DHM or FICZ treatment was determined by immunofluorescence staining. Nuclei were counterstained with DAPI (blue). Data were expressed as means ± SEM. The experiments were replicated 2–3 times. The data shown were from one representative experiment, * *p* < 0.05, ** *p* < 0.01 and, *** *p* < 0.001, **** *p* < 0.0001 and ns means not significant data. Multiple groups were tested by one-way ANOVA followed by Bonferroni’s post hoc test.

**Figure 6 ijms-24-00221-f006:**
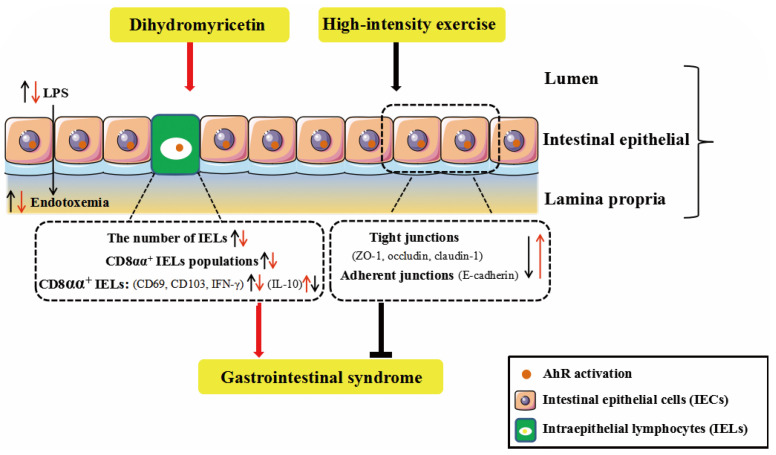
DHM attenuates HIE-induced intestinal barrier dysfunction associated with the modulation of the quantity and phenotype of IELs. The HIE induces an increased intestinal permeability and endotoxemia associated with impaired epithelial barrier, as indicated by the reduced expressions of TJ proteins, which could be ameliorated by DHM supplementation. Furthermore, DHM administration notably suppressed HIE-induced quantitative and phenotypic changes in IELs, particularly CD8αα^+^ IELs and the associated immune markers expression, which might be mediated through the activation of AhR.

**Table 1 ijms-24-00221-t001:** Primers Used for Quantitative Real-Time Polymerase Chain Reaction.

Genes	Forward (5′→3′)	Reverse (5′→3′)
β-actin	CTACCTCATGAAGATCCTGACC	CACAGCTTCTCTTTGATGTCAC
ZO-1	CTGGTGAAGTCTCGGAAAAATG	CATCTCTTGCTGCCAAACTATC
Occludin	TGCTTCATCGCTTCCTTAGTAA	GGGTTCACTCCCATTATGTACA
TNF-a	ATGTCTCAGCCTCTTCTCATTC	GCTTGTCACTCGAATTTTGAGA
IFN-γ	CTTGAAAGACAATCAGGCCATC	CTTGGCAATACTCATGAATGCA
IL-10	ACATTTAGAGACTTGCTCTTGCAC	CTGAGCCAGGCATGATGGAG
IL-15	TCTCCTGGAATTGCAGGTTATT	GCCAGATTCTGCTACATTCTTG
IL-7	GGAAGCTGCTTTTCTAAATCGT	TGTGCCTTGTGATACTGTTAGT
CD132	CAGTGCGAATGAAGACATCAAA	GGAGAACAAATAGTGACTGCAC
CD103	GTACATCTACAACGGACACTCA	GGGGTAAAGGTCATAGATACGG
KGF-1	TGGGCACTATATCTCTAGCTTGC	GGGTGCGACAGAACAGTCT
AhR	CATCGACATAACGGACGAAATC	CTGTTGCTGTTGCTCTAGTTG
CYP1A1 (mouse)	ACCCTTACAAGTATTTGGTCGT	GTCATCATGGTCATAACGTTGG
CYP1A1 (human)	CGTTGTGTCTTTGTAAACCAGT	ACTTAACACCTTGTCGATAGCA
β-actin (human)	CTACCTCATGAAGATCCTCACC	AGTTGAAGGTAGTTTCGTGGAT

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
