# Peer review of "Dihydromyricetin Attenuates High-Intensity Exercise-Induced Intestinal Barrier Dysfunction Associated with the Modulation of the Phenotype of Intestinal Intraepithelial Lymphocytes"

_ijms, 2022, doi:10.3390/ijms24010221_

Round 1
Reviewer 1 Report
I carefully read the article and the work is written in great detail. Congratulations to the authors. This study, which has the original topic title, is a study that contributes to the field.
Reviewer 2 Report
This manuscript has evaluated the effects of the anti-inflammatory agent dihydromyricetin (DHM) on exercise-induced gastrointestinal syndrome (GIS) in male mice. Results showed that DHM partially prevents the generation of GIS, which was associated with improved intestinal barrier integrity and alterations in distinct subsets of intestinal intraepithelial lymphocytes. Consistent with prior studies, DHM was able to activate the Ahr in HEK293 kidney cells, suggesting a mechanistic link between the capacity of DHM to prevent GIS and its capacity to activate the Ahr. From these findings, the authors suggest DHM as a potential therapeutic to prevent GIS in human patients.
Specific comments:
1. For all figure legends, please indicate how many times the experiment was performed, how many mice were included in each experiment, and whether the data shown are from one representative experiment or from multiple combined experiments.
2. For Figure 3A-D: Is it possible to also include absolute numbers of cells, which is often more meaningful than percentages?
3. Section 2.5: Although the data provide strong support that DHM activates the AhR, they do not directly demonstrate that the effects of DHM on GIS are mediated via AhR. The latter would require studies with AhR-deficient animals. As such, this part of the text (especially the title of the section and the figure legend) should be rephrased to indicate that the findings “suggest” a mechanistic link between the effects of DHM and its capacity to activate AhR. Additionally, some prior studies have already provided evidence for the capacity of DHM to activate the AhR, which should be noted in the text with appropriate references.
4. Although the manuscript is assembled and written in a logical manner, the writing can be substantially improved, ideally by professional editing.
Reviewer 3 Report
The study is fine, methods are written in detail, data are convincing, article reads good throughout the manuscript, but only few corrections needed. There were few grammatical mistakes, and a few sentences formation has some flaws.
There are few corrections which can be easily corrected-
Line 35 – consider ‘disclosing’ instead of disclose and ‘includes’ instead of including for better sentence formations.
Line 36-37 – reconsider rewriting the sentence starting for and…
Line 38-40- consider rewriting the sentence.
Line 65- ‘have been proven to be’ instead of have been the proved.
Were all the mice in HIE and HIE+DHM group able to sustain 1 hour running at the speed of 25 m/min every day in the 2nd week to reach its VO2 max?
The acclimatization was only for 10 min at 15 m/ min in the 1st week, which seems not enough to achieve the required vo2 max?
The study by Pengfei Hou et al seems significant enough to determine whether DHM supplementation could diminish HIE-induced intestinal mucosal barrier dysfunction and to uncover the involvement of IELs in DHM-induced benefits.
The materials and methods section of the manuscript is written in detail, which improved the quality of the manuscript.
